# “Not All Who Wander Are Lost”: The Life Transitions and Associated Welfare of Pack Mules Walking the Trails in the Mountainous Gorkha Region, Nepal

**DOI:** 10.3390/ani12223152

**Published:** 2022-11-15

**Authors:** Tamlin Watson, Laura M. Kubasiewicz, Caroline Nye, Sajana Thapa, Stuart L. Norris, Natasha Chamberlain, Faith A. Burden

**Affiliations:** 1The Donkey Sanctuary, Sidmouth EX10 0NU, UK; 2Centre for Rural Policy Research, University of Exeter, Exeter EX4 4PJ, UK; 3Animal Nepal, Dhobhighat, Lalitpur 44600, Nepal

**Keywords:** working equids, life cycle, equid welfare, equid behaviour, translocation, legislation, transportation, monsoon, equid training, biosecurity

## Abstract

**Simple Summary:**

There is now some understanding of the immediate welfare concerns of pack mules in Nepal but there is scant understanding about how they arrive in Nepal, and what challenges they face when they begin work. Using mixed methods, we investigate the owner’s perspectives and using EARS welfare assessments develop a picture about how life is for the mules in Nepal. Mules endure the translocation of many hundreds of kilometres in overloaded vehicles to work in a transient industry where owners often have a lack of experience and understanding of their needs, and where the mules face multiple owners, have a high risk of mortality and, as such, undergo replacement with regularity. This study gives an indication of where potential interventions may have the greatest effect on improving pack mule welfare in Nepal, where improved transport checks and legislation, and support of owners to understand and appreciate their mules’ needs could bring some much-needed relief to this struggling population.

**Abstract:**

Equids in general experience transient lives where ownership may change multiple times, for working equids this can be more extreme where ownership changes are not only numerous but abrupt, and situations encountered prove difficult, diverse and tough for equids to adapt. In this study, we investigate the life cycle of pack mules in Nepal, investigating the challenges they face during their lives through to end of life. To gain insight into the lives of mules, we conducted semi-structured interviews and livelihood surveys with 27 key informants, gathering the perspectives of the people working with mules. Welfare assessments of the mules were undertaken via the Equid Assessment Research and Scoping tool (EARS) by a trained assessor. Mules had to adapt swiftly to changes in industry type, enduring long distance transportation in overloaded vehicles and across country borders with no checks for biosecurity or welfare. Mules had to show swift adaptation to their new environment, to respond to and learn new tasks via inhumanely administered training, using inappropriate techniques, delivered by owners lacking in understanding of mule behaviour and learning. Environmental conditions were often hard; the negotiation of difficult terrain and challenging weather conditions during monsoon and subsequent high-altitude working without acclimatisation likely pushed mules to their biological limits. This study investigates the lives of a population of mules in the mountains of Nepal, developing a better understanding of their needs and their ‘truth’ or ‘telos’ informing what measures will help them to thrive.

## 1. Introduction

Regardless of the thousands of years of domestication equids have retained much of their natural behavioural repertoire, and as such human–equid interactions are still potentially dangerous [1,2] and reliant on some understanding of equid ethology, intelligible communication and the development of trust between the humans who wish to utilise the equid and the equid themselves [3]. Stressful or fear-inducing interactions or experiences increase the likelihood of injury and inhibit the learning necessary for equids to adapt to their circumstances [4]. The human–equid relationship has significant influence on the care equids receive and the decisions made with regards their treatment at key stages within their lives; as such it has a bearing on the subjective perception an equid has of its environment and the actors contained within it, and, therefore, is a fundamental component in securing their long-term welfare [2,5,6,7,8].

Equids often face transient lives where stability of long term ownership can be elusive, and their training, handling and care substandard [9]. Equids are often viewed as commodities that can be bought and sold on a whim; the stability of their existence is heavily dependent on the equid–owner relationship where factors such as incompatibility, conflicting expectations, health issues, and unacceptable behaviours affect the long-term welfare of the equid [10,11,12,13]. The frequent changing of handlers, companions and environments require equid resilience and adaption to changes that are beyond their control but can affect their behaviour, health and welfare. These stressors are just as applicable to equids in high-income countries (HICs) as low-to-middle-income countries (LMICs).

Maintenance of healthy and ultimately productive working equids requires owners to understand and implement suitable management and husbandry throughout the equids’ lifetimes, including adequate nutrition, health care, humane handling, sympathetic training and suitable harnessing. Every interaction and experience an equid receives throughout its life leads to a negative or positive response impacting the emotions, physiology, behaviour and ultimately the welfare of that animal [14]. Anticipation and forward planning are essential to ensure care throughout changing seasons, particularly in regions where extremes of weather, from monsoon to drought, can limit access to adequate supplies and create a subsequent decline in animal welfare [15]. Forcing equids to work through these extremes of temperature and precipitation can cause fatigue, heat or cold stress, reduce performance and result in poor health outcomes [15,16,17].

Investigating in-depth, the lives of a particular animal or population of animals, develops better understanding about what it feels like to be that animal living that life. Ultimately this leads to understanding the needs of that animal and its ‘truth’ or ‘telos’, thus informing what measures will help that animal to thrive [18,19]. Mountainous regions in Nepal cover over half its total land mass and communities here are usually the poorest in the country. There is a heavy reliance on animal transportation and distribution in the absence of any alternative able to access these remote areas [20]. Mules in this region have usually worked for at least one season in the brick kilns of India; following this induction into work they travel into neighbouring countries and at this point may change industry (TDS, unpublished data in 2019). This manuscript aims to investigate the lives of equids walking the trails in Gorkha, Nepal; documenting and discussing their transient existence and the challenges they face to gain understanding and enable purchase on their ‘telos’.

## 2. Materials and Methods

Using a mixed methods approach this investigation delved into mule life transitions and was centred around equid owner/driver semi-structured interviews (SSIs), livelihood surveys, and equid welfare assessments using the Equid Assessment, Research and Scoping (EARS) tool [21]. Livelihoods surveys, SSIs and welfare assessments were undertaken whilst owners and mules were at rest between work periods. Livelihood survey and welfare assessment data were recorded into Open Data Kit Collect (ODK Collect) [22] on digital equipment, and uploaded to a UK server when in range of an internet connection.

### 2.1. Study Sites

Fieldwork was conducted in Gorkha, Nepal from the 12 November to 25 November 2018. A partner organisation Animal Nepal (AN) provided logistical support as they had a good knowledge of the study area and the mule populations, and could provide interpreters able to communicate in local dialects and fluent in English. Observations and local knowledge were shared between the research team, which included a veterinarian from AN, during fieldwork and either audio recorded or logged in field notes.

The Gorkha study sites included the habitations of Arkhet Bazar, Soti Khola, Maccha Khola, and Tatopani. Gorkha is a mountainous region mainly connected via narrow mountain trails; most villages were accessible only on foot, with the exception of Arkhet Bazar and Soti Khola, although access to these were limited to off-road vehicles and some trucks/buses.

### 2.2. Participant Recruitment

We secured participants on an ad hoc basis. Interactions with people and mules were not time-limited unless owners/drivers had to leave for work. Human populations are low density and dispersed in the region, so the number of suitable participants was limited.

Interviews were conducted on an individual basis apart from one mule owner/driver group who were interviewed together. Participation was voluntary, unpaid, and limited to adults over 18 years old who owned, worked with or traded mules. Consent was obtained verbally and audio-recorded. All participants received a unique code to ensure data were pseudo-anonymised. Participants were provided with contact details for AN and given the right to withdraw their data within two weeks of interview.

### 2.3. Topographic Map

A topographic map of the mule journey was developed using QGIS [23,24,25,26] and was based on the likely route roughly described by the mule trader which was confirmed by professionals from AN.

### 2.4. Data Collection

#### 2.4.1. Quantitative Data—Livelihood Surveys

Each survey was recorded electronically on digital equipment using an ODK Collect [22] form containing pre-set questions (see Appendix A). Questions recorded demographic information (age, gender, ethnic group, religion, job role, income) and information about mule ownership (number of mules owned, duration of ownership, health issues of concern to owners). The study forms part of a wider study so only the information pertinent to this article has been included.

#### 2.4.2. Quantitative Data—EARS Assessments

The EARS assessments formed part of a wider project so only age composition and behavioural data were included for the purposes of this article (see Appendix A for a list of EARS scoping protocol questions). All of the assessed equids were mules, with no other equid hybrids or species seen in the region. Not all owners had mules available for assessment as they either had removed them for work or had released them to graze elsewhere. Welfare assessments on working mules were conducted by a trained assessor (TW) following EARS methodology detailed in Raw, et al. [21].

#### 2.4.3. Qualitative Data—Semi-Structured Interviews

Twenty-seven semi-structured interviews (SSIs) with mule owners and those working with mules were undertaken by TW and LMK. Each SSI lasted between 20 and 54 min; core questions in pre-determined themes formed the initial foundation of interviews. As participants relaxed, unexpected themes emerged inductively (see Appendix A for a list of core questions). Questions were given in English by LMK and TW and were translated into Nepalese via Nepalese interpreters from AN. Interviews were recorded by Dictaphone and later transcribed. Using the software package NVivo (NVivo 12 qualitative data analysis software, V.12.5.0, QSR International), qualitative data were uploaded and analysed. An iterative inductive approach allowed analysis and identification of new emerging themes. For this article, themes were included concerning translocation/transportation, adjustment to a new environment and training, and management and hazards during natural weather events.

### 2.5. Ethics

The study and protocols herein were conducted in accordance with the Declaration of Helsinki [18] and was approved by the Ethics Committee of The Donkey Sanctuary, UK, with project Number 2019-AIM2-NEPAL.

Recruitment of participants was on a voluntary basis. Informed consent was gained from each participant and audio recorded prior to inclusion. All participants and location details were stored; all data were anonymised. All participants were given the right to withdraw within a two-week period by contacting a member of the Animal Nepal team. No participants withdrew. All mules were welfare assessed using non-invasive techniques throughout.

## 3. Results

### 3.1. Quantitative Data

The number of interviews and livelihood surveys reflect the low population density of the mules and their owners. As this is a study into a distinct population within the Nepal mountains, and as the sample size is limited, the results cannot be generalised more widely without further investigations of mule populations elsewhere.

#### 3.1.1. Livelihoods Survey

Twenty-seven surveys from owners and those working with mules were included in this study; recording twenty-four mule owners/drivers, one mule trader, and two veterinary technicians. Participants included members of the community directly linked to the mules through present ownership, or those working directly with the mules either as their main job role or as part of their job role.

#### 3.1.2. EARS Welfare Assessments

The assessed population consisted of 156 geldings and 10 stallions, belonging to 17 owners. One specific cohort of fourteen mules assessed belonged to a single mule trader who had transported twenty-three mules over the border from India; during interactions with the mule trader, he reduced this number to eighteen as he may have realised the illegality of transporting so many mules in one vehicle; however, eighteen mules is still overloaded. The mules worked as pack animals for the distribution of heavy goods such as rice, oil and building supplies to households and businesses; mules did not perform any other roles.

When the assessor approached the mules, 49% (*n* = 81) exhibited negative behavioural responses such as aggression, head shyness, or had unpredictable/startle reactions. The remaining 51% (*n* = 85) exhibited friendly responses, showing interest in interacting such as turning their head or moving towards the assessor, although 4% (*n* = 6) become less relaxed during the assessment where their behaviour became unpredictable and startle responses increased. Two percent (*n* = 3) of the mules exhibited apathetic behaviour. All mules had other obvious ailments including open wounds, nasal and eye discharge; one of these mules was worked despite having a broken hind leg and open wounds infested with flies.

Approximately half the mules (54% *n* = 88) could not be aged, being either removed by their owner to return to work or because of exhibiting negative behaviour such as nervousness or aggression such that the assessor was unable to examine the individual. The age composition of mules that could be assessed were 44% (*n* = 34) above 5 years old, 24% (*n* = 19) under 5 but over 3 years old, and 31% (*n* = 24) under 3 but over 1 year old. One mule was over 20 years old.

### 3.2. Qualitative Data

#### 3.2.1. Transitions: Translocation/Transportation to Nepal

Working mules are a lifeline to the communities in Gorkha region, Nepal, where steep valleys and mountains are inaccessible by motorised vehicle. Before mules’ presence in the region people had to rely on carrying all the goods using human power alone, often walking considerable distances to obtain the goods they needed. According to one owner, mules arrived and began being used for work in Gorkha within the last 12 years, and since this time have been regularly transported across the border from India by traders. A mule trader we interviewed had just arrived from Lucknow, India, bringing with him a lorry overloaded with 23 large mules (14–14.2 hands high or 1.42–1.44 m); according to Indian legislation loading should be restricted to between four and six mules depending on the size of vehicle [27]. According to the mule trader the common mule transportation route from Lucknow crosses the border at Nepalgunj, continues onwards through Nepal to Pokhara then, for livestock containing vehicles, Gorkha Bazar is the final destination where livestock is generally offloaded. After Gorkha, roads deteriorate to dirt tracks until arriving at Arkhet Bazar, a small village marking the start of the Manaslu circuit (and the end of most other vehicular access); approximately 672 km in total (see Figure 1). All mules working in the region came from India; there does not appear to be a breeding strategy for working mules in Nepal (field notes, TDS, 2018). After arrival in Nepal, they work either within the brick kiln industry or in goods distribution in remote mountainous regions, such as Gorkha. Mules that are experienced and ‘broken’ into work are more attractive to those wishing to purchase equids for work as they adapt more readily, this was confirmed in an interview with a mule trader who was in the process of selling mules in Soti Khola:

They [mules] have been working in India; in brick kilns […] if they [mules] work in India, then it is easier for them to work here.(Mule trader)

Indian traders generally buy mules, either from Barabanki, India (there is a large equine fair there), from owners working in the brick kilns, or some kilns loan (for payment) mules to owners for a season before they are translocated elsewhere (TDS, unpublished data in 2019). The trader we spoke to bought 23 mules for INR 58,000 each (approximately GBP 393.00/USD 456.00) from people in the brick kilns and hoped to sell them for NPR 70,550 Nepalese rupees (GBP 477.00/USD 555.00). If unsuccessful selling all the mules “within a fortnight” he would transport unsold mules back to India to continue work in the kilns.

We asked the mule trader to describe the 672-kilometre journey from his starting point at Krishna Nagar on the west side of Lucknow (Uttar Pradesh, India) to Gorkha Bazar (Gorkha, Nepal) (Figure 1):

Just [one break in the journey] in Gorkha [Bazar] while we are coming to Gorkha [Bazar] there is a river and we stop them to have some water and food [unclear whether the food and water is for the mules or the trader].(Mule trader)

This appears to be the only break in their ~20 h journey, which is a long time for a lorry (particularly an overloaded one) to be transporting animals, especially on such uneven and difficult road terrain. As there was only one break in this journey, no checks could have occurred at the border, or if they did, the mules remained on-board. After unloading at Gorkha Bazar, the mules walked for “two hours” to their final destination as the roads become too difficult to traverse in a livestock-transporting vehicle. However, this would be an under estimation (possibly for our benefit as the trader was probably aware of the excessive duration of journey and the illegality of some aspects of it) and 9–10 h walking is more accurate.

Walking mules, sometimes to avoid driving through border checks, is a biosecurity concern mentioned by a veterinarian within the research team and audio recorded after we had interviewed the trader.

They bring in glanders [Glanders is a notifiable infectious disease mainly affecting Equidae, though other species, including humans, can get the disease [28]] the disease. The government restricted the entry of mules to this particular state (during glanders outbreaks) from Uttar Pradesh, so they [the traders and migratory workers with mules] walked [to avoid detection]. They come in somehow, we don’t know how, illegally, but we don’t know how they get here.(Veterinarian AN)

#### 3.2.2. Transitions: Adaptation to New Surroundings, Handling and Training

Prospective owners assumed mules arriving were work-ready due to the mules having already worked in brick kilns in India; the mules’ previous work experience kept the time required for training them short, although familiarising them with the new working regime was not without its issues:

[…] they do not know the working route—they are not habituated initially so three or four fell down on the route.(Mule owner)

The mules need to adjust to the new environmental working conditions when carrying loads in the mountains (as opposed to the brick kilns) as the terrain is more varied, having narrow suspension bridges, rivers and steeps slopes for the mules to negotiate:

Yeah they needed training because there is a vast difference between carrying loads in the brick factory and carrying loads over here. In the brick kiln, they only work for five to ten minute distance, and they have a sack in both side like a bag, so it is easier for them. But putting loads on the [pack] saddle it’s a different thing and tying them and walking in a very narrow road, it’s very different so need to be with two or three person [guiding them] before otherwise it will be very hard for them to carry over here.(Mule owner)

In brick kilns, working mules have a pannier attached either side of their body, are expected to walk the same route back and forth between brick stacks and kilns, over relatively even ground, from 15 to 40 times a day usually on a reasonably level surface (TDS, unpublished data in 2018). Contrary to work in the mountains where they walk for hours to different locations (dependent on demand) on challenging terrain with their loads strapped to their packsaddle:

It takes almost one month for them to train and they need to learn how to work the curved routes [winding trails], and they need to know where to jump, how to cross the small rivers and to learn how to carry loads, they jump a lot when we first load them.(Mule owner)

Training to walk over suspension bridges seems a particularly difficult process for the mules; this process involves being pushed or dragged by 3–4 people, or mules are tied to other mules and urged on by their owners. One owner said their method to make them cross the bridges initially was “dragging, slowly and slowly dragging” the mules across by attaching them to other mules. Additionally, when working out carrying capacity for each mule, it seems to be rather ad hoc:

If a mule sweats and cannot walk the load is too heavy.(Mule owner)

Some owners stated they load new or younger mules with lighter loads, although there was no indication of how much lighter or how long they implemented this method. There are significant adaptations required as the mules adjust to their new home, companions, handlers, routine, and food that may not only affect stress levels, behaviour and health but also increase mortality.

#### 3.2.3. Transitions: Adjustment to Environmental Conditions

Nepal is a land of extremes, from hot, dry conditions, to the torrential rain experienced during monsoon season. These extremes put pressure on owners when providing for their mules to ensure their equid management changes in response to the environmental conditions:

During the monsoons you need to take care of the mules with extra things, it’s just like humans we need good food after work so I give them good food […]. You have to take care of their feet properly during the monsoon, if you do that your mules are going to be alright in every season […], even during the cold season I give them honey and eggs. When you reach up there [further up into the mountain ranges] in the cold I give them Kukuri [a Nepalese brand] rum to save them from the cold.(Mule owner)

An owner also provided shelter to mitigate some of the issues, but most owners leave their mules out in the rain or continue to work them:

I can’t let my animals graze up there because of the [potential for] landslide and even down there [lower in the valley] the rivers will grow so what I do is I just make a rope here and pull around the tent so they can be safe. I save my animals like that but some other mule owners they let them [loose] wherever they are and so that’s the reason their feet are under the ground in all the mud and even their body are all wet so they get sick very frequently […]. In my opinion, we shouldn’t use our animals because the feet get hurt most at that time and most of them limp so from this monsoon, I try to decrease my movement of my animals to save them […].(Mule owner)

Owners admitted to losing mules in floods and there was a very real threat to human life:

It’s very dangerous during the monsoons, so when you work [the mules] through there [monsoon] I think I might die and my mules might die I keep on thinking that, it’s very, very difficult.(Mule owner)

Lack of money and demand for goods meant owners often had to work during monsoon as they and their animals needed food and owners had no earnings otherwise. They were aware that loading their mules while wet was inadvisable, but many felt that throwing a plastic sheet over their mules was sufficient to overcome any issues. The scarring seen on almost all the mules had its origin from mules working whilst wet during the monsoon; one owner admitted the mules had ‘lots of wounds on their backs during the rainy season’. To mitigate the problem some owners tried to work mules further up into the mountains where the rain was less persistent. Mules distributed goods high up in the Larkye pass (5106 m)—a seven-day trip, and transported cement and other supplies for the building of a hydroelectric power station at the head of the Tsum valley (5093 m). If the rains were heavy when they were trying to ascend or return from the mountains, they would change their plans:

[When there is bad weather] it affects us a lot because if we are going up and its rained we have to stop there, eat there, stay there […]. If there is no snow they can go up, but cannot go to Larkye Pass and Tibet […] as we cannot reach to our end place [destination] we have to leave everything in between the way [at a location on the trail set up for storing goods].(Mule owner)

These high-altitude treks were occasional, sometimes only two to three times per year, leading to many mules becoming sick or dying from the effects of working at altitude without acclimatisation, according to their owners. People tried various methods to treat altitude sickness in their mules ranging from allopathic medicine or traditional remedies, but mules that did not respond to treatment invariably died:

So when we are up there [at high altitude] it [mule] show the signs of headache and some walk very fast when they have a headache and some people cannot work [the people also get altitude sickness], then we can tell they have an altitude sickness or something like. Somebody get medicine in some animals it works, in some it doesn’t and they die […] so we try to treat it, we give them garlic and hot water and some alcohol.(Mule owner)

## 4. Discussion

### 4.1. Translocation and Transportation

The mule trader traced his 672 km, approximately 20 h duration journey, to its origins in Lucknow, India. Equids transported in groups tend to be those of lower value in higher income countries [29], but it is the norm in LMICs. The mules would doubtless have been loaded without any training or preparation for the transportation, and although suboptimal travelling conditions can be tolerated for short journeys (up to around 3 h) [29]. Mules preparing for loading and the subsequent transportation would experience enforced separation from familiar environments, social regroupings, mixing of unfamiliar conspecifics, high stocking densities, inadequate vehicle provision, inappropriate handling and driving methods, water and feed deprivation; all factors known to increase risk of disease transmission, cause stress, anxiety and associated behavioural manifestations [30].

The ethology of equids, including ‘innate neophobia’ and aversion to confinement, would exacerbate the stress of transportation [31]. Their behavioural opportunities and control are limited, particularly when transported in overloaded vehicles lacking in adequate ventilation, and in the challenging environmental conditions of India and Nepal’s climate where high temperature and humidity could induce thermal stress [32]. Mules who had experienced previous aversive transportation and handling, which would have been all the mules in this study, may exhibit increased problem behaviours before loading and would be less resilient to cope with the stresses this process presents [33]. The mule trader expressly mentioned he had travelled the animals with only one rest break which occurred when the mules were unloaded prior to walking the remaining distance (9–10 h) to their destination. It was unclear whether the rest break was actually benefitting the equids or just the drivers; no feed or water were offered to the mules during the vehicular part of the journey.

In India, although legislation specifically mentions equids, enforcement is rarely, if ever, implemented and fines are very low [34]. The Prevention of Cruelty Against Animals Act, 1960 prohibits the transportation of any animals which causes suffering such as overcrowding, forcing animals to walk long distances on foot, loading without ramps, activities punishable with a fine and/up to three months in jail [35]. The Transport of Animals Rules, 1978 [36], underwrites that equids are required to be transported with veterinary certification, vaccinations, with a maximum of four to six equids properly partitioned, and, with a recent amendment to The Motor Vehicles (Amendment) Act, 2015 [27], be afforded 2.25 m^2^ per equid. There are no definitive conclusions in scientific literature as to the exact spatial requirements for equids being transported although there are some minimum stipulations. Equids need to be able to spread their legs and to raise and lower their heads for balance during transportation. Experts suggest equids should have enough space to adopt a balanced position where there is a free space of a minimum of 20 cm all around the equid’s body [37], a minimum allowance of 1.75 m^2^ per adult horse or 1 m^2^ per adult pony [38] or equids should travel in a 1.9 m^2^ bay [39]. The mules in this study were adult horse and pony size, so we include both allowances here. This is in stark contrast to the mules seen in this study that travelled as a group of 23 transported by a mule trader who had travelled from Uttar Pradesh in India; mules would have had no space at all around their bodies, thus limiting their ability to regulate their body heat, to breathe easily and to balance during travel. Possibly, upon realisation we were interested in the mules’ welfare he reduced the number of mules he transported to 18, which is still far greater than is legally allowed. Transportation in higher stocking densities is known to lead to poor welfare through injuries, physiological stress, heat stress, sensory overstimulation, resting problems, restriction of movement, social stressors such as aggression, and poor air quality [37,40].

In Nepal, equids are not mentioned specifically within their Animal Health and Livestock Services Act 2055 (1999) [41]. The legislation does stipulate that any animals transported should not be confined in the vehicle for more than eight consecutive hours without rest, food and water and that those being imported should go through a quarantine checkpoint before being allowed entry over the border from India. In this study, the driver clearly did not abide by Nepalese transportation legislation as mules were given no rest, food or water whilst travelling on the vehicle, nor were the mules unloaded after 8 h. Reports written regarding the translocation of equids between India and Nepal noted that facilities at the open-border checkpoint were inadequate and animals were neither removed for inspection or held if found to be diseased [42] and smuggling animals across the border was prolific [43]. This was supported by information the authors repeatedly received whilst conducting fieldwork in the brick kilns in Gujarat, India; vehicles were rarely, if ever, stopped at checkpoints and if they were stopped officials would often accept monetary encouragement to allow their vehicle passage and penalties were so low as to be no deterrent (TDS, unpublished data on May 2018). The lack of checks at the India/Nepal border had impacts on biosecurity during the glanders outbreak in India [44]; economic migrant workers and their equids were illegally transported across the border to begin work in brick kilns after receiving payment in advance from brick kiln owners to secure their labour the previous working season [45].

Dehydration and risk to respiratory health are common outcomes for equids when transported, particularly if journeys are long, and have challenging environmental conditions [29]. Dehydration can be induced from lack of water provision and anxiety-derived diarrhoea, particularly if a vehicle is densely loaded in hot and humid conditions [46]. In turn, this may affect metabolic processes potentially inducing laminitis, reducing renal function and increasing the chances of colon impaction, both during and after the transportation has ended [47,48].To avoid this scenario, water should be offered every 2–4 h, particularly when the weather is hot and humid [49]. As would have been the case in India and Nepal, however, the mules received no water for the duration of the journey until unloaded. Lack of adequate ventilation and heat stress in an overloaded vehicle would have added to the trauma these mules were experiencing, swiftly rendering the limited airflow contaminated. Over time, this would have increased carbon dioxide and ammonia concentrations [50], escalating bacteria and endotoxins, and thus amplifying the risk of airway inflammation, bacterial pneumonia and pleuropneumonia infections [46,51]. To mitigate against these issues vehicle design needs to incorporate adequate ventilation and thermal insulation, be thoroughly cleaned and disinfected between uses, transportation of equids from different locations limited during the same journey and equids travelled in low densities, preferably during cooler conditions [46]. Stress from transportation causes the activation of hormones which in turn, raise heart and respiratory rates, sweating and defecation increasing the already hot and contaminated vehicle whilst also influencing the equid’s immune response leaving them more susceptible to disease [37,46,52].

According to Phelps and LeDoux [53], short-term fear memories, such as those which were likely induced on this journey, are easily transformed into long term memories because of the important survival information they may convey about danger. Depending on the temperament of the animal future responses to a similar fear-inducing stimulus may not manifest as external behavioural responses or may manifest as responses handlers may not associate with fear such as tail swishing, sweating, head raising, showing the whites of eyes or aggression [54]. Aggression typically escalates inhumane responses from human handlers to increasing fearfulness or frustration with the animals, whilst not comprehending that the animals too are fearful [55]. A vicious cycle. Working equids in India and Nepal experience transportation throughout their lives. Initially being delivered to be sold in markets and fairs, then with owners travelling for migratory work to brick kilns [45,56,57] or when translocated many hundreds of miles for work at pilgrimage sites or across country borders, such as is the case in this study.

Injuries are common during transport particularly when travelling with inadequately prepared handlers in vehicles ill equipped for equid transportation, and where there is mixing of unfamiliar or poorly matched equids which have little or no experience of the stimuli they will encounter throughout the entire handling, loading and travelling experience [29]. Injuries during transportation link to various factors. Human factors such as inappropriate or coercive handling during loading and unloading, mixing unfamiliar or aggressive equids in high densities can all increase stress and anxiety in equids and escalate transport related problem behaviours (TRPBs) such as aggression, and rapid avoidance movements leading to leg or body injuries from slipping or kicking [58,59]. Vehicle factors such as a lack of vehicle checks, poor braking systems, unsafe and unsuitable vehicles can lead to abrasions, leg wounds, and fractures or joint dislocations when equids are restrained and drivers use excessive braking whilst driving [60]. Simple measures can make equid safety during transportation easily achievable, ensure vehicles are fit for purpose with protective features such as padding, load to recommended stocking density, remove aggressive individuals, ensure drivers are competent to drive livestock and add bedding to soak up urine and reduce slipping [58,61]. Habituation to transportation is an ideal method for preparing livestock for transportation [33,62] but is likely widely overlooked by those under pressure to move livestock around for a living. The mules would have travelled in a vehicle sub-optimal for transportation of mules, more likely designed for other livestock such as buffalo, and was definitely overstocked; although there is no stipulation within Nepalese legislation regarding the transportation of mules, the legal requirement for buffalo transportation is a maximum of 15 animals [63,64]. High stocking densities create difficulties for animals to maintain posture during transport due to limited available space to realign themselves [40] increasing the likelihood of injuries and stress, particularly as mules in this study may have been unfamiliar to each other. On disembarkation from the lorry, the mules then endured a 9–10 h journey walking to their final destination. Although there are no specific legislative rules in Nepal regarding walking livestock, the walking of these mules for such a long distance particularly after a long journey would have affected the welfare of these animals. In India, The Prevention of Cruelty to Animals (Transport of Animals on Foot) Rules 2001 [65] restrict the transport of animals on foot when the “distance from the boundary of village or town or city of the origin of such transport to the last destination is 5 km or more than 5 km”.

Lack of resource allocation—both financial and human—inhibits effective legislative monitoring and enforcement, leading to the continued flouting of transport and animal welfare laws within both India and Nepal. To improve the long-term behaviour and welfare of working equids whilst being transported inclusion of regular rest stops to provide food and water, adequate ventilation and sufficient space would ensure mules were transported with as little stress as possible, this clearly did not occur for the mules in this study.

### 4.2. Adaptation to New Surroundings, Handling and Training

Fifty-five percent *(n* = 43) of mules assessed were under five years old, and over half of these were under three years old. The worrying trend within the most recently translocated equids belonging to the mule trader was their young age; out of fourteen animals assessed, all were between 1 and 3 years old, apart from one that was over 5. All equids, where their origins were known in the study, had been carrying loads within brick kilns in India for a whole six-month season before being transported across the border to Nepal, which is known to be the usual outcome for these equids [66]. Putting equids to work at such a young age contravenes recommended government guidance in Nepal where equids are recommended to be a minimum of three years old to begin work [67]. This is still younger than other recommendations where equids deemed suitably developed and fit for work are aged four years and older [15,68,69]. Although equids may appear mature at two years old, putting any equids to work before their bodies are mature (at ideally 5–6 years old), puts strain on their bones and joints [57,68]. This excessive loading results in gait and hoof abnormalities, joint and other load related injuries, ataxia, and results in negative behaviour patterns such as aggression, avoidance and apathy [70,71].

Livestock are an essential resource for remote socio-economically marginalised mountain communities in Nepal, where economic vulnerability [72] means owners may be forced to put mules to work before they are fully mature. Although working equids at an early age causes long-term damage, having equids proven as fit for work and already broken to harness increases their value and attractiveness for communities who have limited time and funds to wait for equids to be ready for work. The mules within this study were broken at a young age in India before translocation to Nepal, where they had to adapt to the new harnessing equipment, handlers and environment. Ideally, training and equipment introduction should be gradual, ensuring familiarity at each stage before equids are asked to carry or pull anything of weight as inappropriate loading of equids is known to cause physiological, musculoskeletal and integumentary trauma [70]. We use the term broken or trained loosely, as methods used in this study to yield equids to harness are often inhumane and ill conceived.

Studies investigating the welfare of working equids rarely investigate training methods, although studies undertaken in India by Watson, et al. [57] reported young equids being harnessed without any form of gradual introduction, often tying them to another equid until they cease struggling; load packing is at full capacity immediately after this point. Within this study, mules trained to take to their new working systems within one month; owners used ad hoc methods to ascertain load capacity and forced mules forwards to make them work the trails. The challenges of a completely different way of working—narrow trails, suspension bridges, steep inclines, new companions and handlers—was given little or no consideration by owners desperate to make their mules work as quickly as possible. Owners openly admitted pushing newly acquired mules to their physiological and mental limits too soon, which often resulted in mule fatalities and risks to human health and safety.

Therefore, humane training appears to be limited. More fundamental to the process is positive punishment (using force to correct), negative reinforcement, flooding and learned helplessness. Learned helplessness occurs when an animal subjected to high levels of an aversive stimulus or numerous stimuli learns they are unable to escape or control the situation [73,74]. All interactions give equids opportunities for learning, so any deficits in handling, training and exposure to stressful situations contribute to conflicted, confused and potentially problematic behavioural responses [75]. This may, in part, explain the mules’ negative responses to observer approach and handling we recorded previously [76]. Unfortunately, the more fearful or aggressive an animal becomes, the more adverse handling may become [10,77] the greater the risk of injury to handlers [78] and the poorer the welfare outcomes for the mules [10]. This creates a cycle of increasingly aversive handling, particularly, if the ‘bond of interdependence is broken’ [13] such as when mules are driven and handled by people other than the owners. Those caring for animals should have an understanding of the unique biological drives and instincts that have evolved to create the individuals in their care [79] and ensure they comprehend their responsibility in ensuring their animals are given the best opportunity to live their life well [18].

### 4.3. Adjustment to Environmental Conditions

Handlers and training practices are only a small element of what mules have to adapt to, in addition, environmental and climactic conditions play a significant factor in their lives. Monsoon conditions disrupt the lives of these mountain communities, trails become dangerous, landslides frequent, and people have to prepare to ensure both themselves and their animals survive. Despite this, ongoing demand for goods throughout monsoon and a pressure to earn money outweighs risks to lives so handlers continue to work. Owners reported that mules suffered open wounds and hoof issues from working during monsoon. Although wounds and scarring may have been the result of ill-fitting and soiled equipment [80], putting equipment and loads on wet animals causes friction and rubbing and eventually skin integumentary trauma; the scarring resulting from equipment was recorded in a previous study [76].

While attempting to avoid the worst of the rains, owners distributed goods further up the trails into the mountains. Although owners lost mules when working them at altitude, they still took the risk in working them above 5000 m without any acclimatisation; altitude sickness usually begins to become an issue at elevations over 2500 m. High altitudes present numerous pressures to animal health because of reduced oxygen availability leading to acute hypoxia for those not already acclimatised [81,82]. To have optimum survival and adaptability at high altitudes, animals (including people) can develop certain physiological adaptations that enable adaptation to this extreme living, but for most species it takes many generations for these adaptive genetic mechanisms to manifest [81,83], it is unknown whether, as a hybrid, it would manifest in mules where the parent species would require heritable dominant traits. Working animals at altitude without adequate time for acclimatisation pushes animals to their physiological limits, even healthy equids become hypoxemic (have low blood oxygen) at high altitude [84]; so equids suffering from stress, over work, wounds, and injuries are likely to be even more susceptible to its effects. Owners, although understanding that working at altitude was causing issues for their mules, resorted to the use of traditional remedies to treat symptoms whilst being aware and resigned to the fact that the in the absence of veterinary support, adequate knowledge and any other alternative that some mules’ reaction to working at altitude would be life limiting. Mule owners do not have the benefit of time and as the mules they purchase would not have worked at altitude there needs to be a realistic solution to enable them to work regularly and more safely. Although there are limited studies regarding the acclimatisation of equids to working at altitude, Greene et al. [85] concluded that, consistent with human studies into this topic, slow and steady acclimatisation over a period of 3–4 days, would give equids the support to acclimate to working at altitude, and lessen the mortality associated with the present working practice.

Increasing intensity and extremes of weather (both wet and dry) are becoming more prevalent with climate change, and as such, monsoon seasons are becoming increasingly challenging as flooding, natural hazards and food insecurity rise [86]. Communities already socioeconomically marginalised have limited livelihood choices, little capacity to adapt, scant access to good agricultural land and other essential services [72,87] leaving them most vulnerable to natural hazards and the impacts of climate change [88]. This has significant implications for mule welfare; lack of owner awareness and increasingly unreliable access to clean water and forage materials may leave mule owners struggling to mitigate this situation, causing risks to the lives, welfare and health of both the owners themselves and the mules they depend upon. To really understand how to ameliorate equid welfare context-specific intervention not only needs to consider the biology, husbandry, development or adaptations influencing welfare but also the socio-economic and environmental constraints of those who care for them [18].

### 4.4. Study Limitations

To provide interpretation and logistical support during fieldwork the organisation Animal Nepal assisted. Animal Nepal provides veterinary interventions to mule owners within the region so acknowledgement is given that this may have influenced interview and survey responses of some participants. Without this assistance, however, language interpretation and travel in the region would have been extremely difficult.

## 5. Conclusions

Understanding the ‘telos’ of mules in Gorkha is acknowledging their nature as prey animals needing support and reassurance, humane handling and to appreciate their physiological and behavioural limitations [18]. Owners showed some understanding of the issues affecting their mules’ welfare, although their limited knowledge of the needs and behaviour of their mules, lack of access to support, knowledgeable guidance and a dearth of monetary funds impeded their ability to care adequately for them. Some elements of the lives of mules in Nepal are beyond the control of their owners, although not beyond the protections afforded by the legal systems in India and Nepal. Transport legislation needs recognition and enforcement; checkpoints need to be functioning and mule traders need encouragement to appreciate the nature, character and welfare of their cargo.

Most people who have experience of animal husbandry appreciate the necessity to understand the physical and psychological needs of their livestock. The lack of experience keeping mules in this region, and the transient nature of their employment may in part explain the limited owner or handler responsibility or attentive care given to these animals, meaning mules rarely live according to their needs and adaptations. To support and improve welfare for these mules, or any population of animals, owners should be fully involved in the conception and development of steps to improve the welfare of their mules, creating a shared vision where the legitimacy of each other’s views are respected and aid the development of a more holistic process for maintaining equid welfare. Interventions should avoid ethnocentricity and bias; welfare practitioners should work towards ‘decolonising their practice’ by understanding and listening to the communities they interact with [89]. Even when owners and handlers are too economically disadvantaged to improve all aspects of their mules’ lives, increasing capacity by supporting owners in learning to understand the needs, abilities and biological drives of their mules may enhance responsibility and empathy towards their equids. Doing so may give the opportunity for mules to live a life more closely matching their needs in the mountains of Nepal.

## Figures and Tables

**Figure 1 animals-12-03152-f001:**
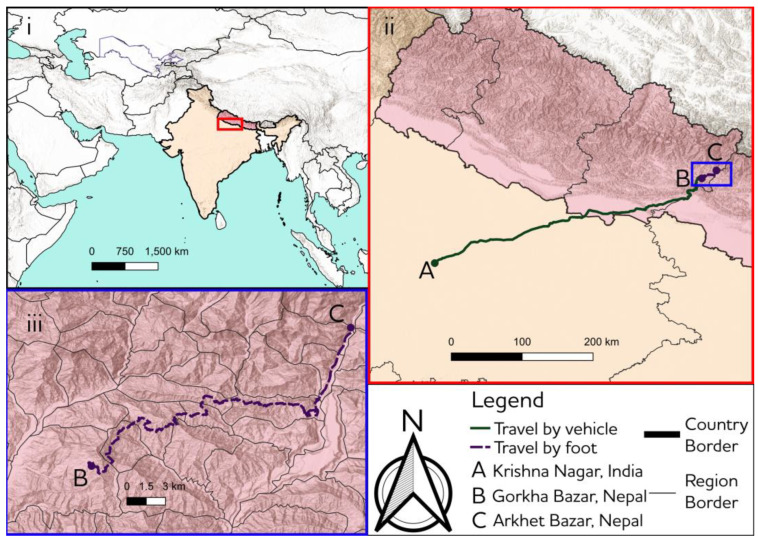
Journey of the mules from Krishna Nagar, India, to Gorkha Bazar, Nepal. i. Map showing location of journey in India and Nepal ii. Main stage of journey using vehicular transportation from India to Nepal iii. Final stage of journey on foot from unloading to ultimate destination in Gorkha. Coordinate reference system EPSG:4326 WGS 84.

## Data Availability

The datasets generated and analysed during this study contain sensitive material as such are not publicly available.

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
