# Peer review of "“Not All Who Wander Are Lost”: The Life Transitions and Associated Welfare of Pack Mules Walking the Trails in the Mountainous Gorkha Region, Nepal"

_animals, 2022, doi:10.3390/ani12223152_

Round 1

Reviewer 1 Report

I read this paper with great interest. It clearly tells the story behind some hard, dedicated work in difficult conditions.  The authors are to be commended for bringing to our attention the plight of pack mules in Nepal.  Lets hope publishing their findings leads to real, long-lasting improvements in pack mule welfare.   

L66 Investigating in-depth the lives of a particular animal or population of animals, develops better understanding about how it is to be that animal living that life, which ultimately helps the understanding of that animal’s needs and their ‘truth’ or ‘telos’ informing what measures will help that animal to thrive [10,11]. Awkward sentence – re-write for clarity (use some  commas).  Can humans ever 'be an animal'? Suggest Investigating in-depth, the life of a particular animal or population of animals, develops better understanding about what it feels like to be that animal living that life.  Ultimately this leads to understanding the needs of that animal and its ‘truth’ or ‘telos’ thus informing what measures will help that animal to thrive.L78 a good life??? Use an alternative adjective to the subjcetive 'good' The mules will never have a good life - perhaps 'a better life' or 'less stressful life'  

L86 ODK Collect???? unexplained abbreviation  Explained L117 move

191 depending on the size if vehicle - spelling 'of'

203 use a unique font to denote quotations. They [mules] have been working in India; in brick kilns […] if they [mules] work in India, then it is easier for them to work here. (Mule trader)

L215 denote quotation

L 447 to get them to work the trailswork or walk???

L510 To really understand how to ameliorate equid welfare. wrong use of ameliorate (to make better). Use an adjective before welfare such as 'substandard' or 'poor quality'

L542 decolonializing their practice.  spelling 'decolonizing' 

L546 By doing so may give the opportunity for mules to live a life more closely matching their needs in the mountains of Nepal.  Suggest re-write to' Doing so may give mules the opportunity to live a life more closely matching their needs in the mountains of Nepal.

Author Response

Dear Reviewer, 

Heartfelt thanks for your constructive suggestions and corrections. We hope you find the edits to your satisfaction, please find attached a document responding to your comments. 

Kind regards

The Authors

Reviewer 2 Report

While this study is a little short on objective measurements, the authors have done an excellent job given the difficult circumstances.  This investigation is a model of what One Health and Veterinary researchers should be pursuing.  Thank you for your efforts. 

Author Response

Dear Reviewer, 

Thank you for such encouragement regarding our manuscript, we really appreciate it and hope it gets the opportunity for dispersal into the wider public arena..

all the best

Tamlin

Reviewer 3 Report

Thank you for a great concept and an excellent article on mule welfare. Please consider the following edits.

Line 45: I would suggest including a paragraph on the importance of mule and human-animal interactions (HAIs) at the start of the introduction.

Line 156-159: It would be worth adding the small sample size limitation, especially for survey-based studies, as you interviewed 27 participants.

Line 137-142: How you analyzed your categorical data obtained through EARS questions?

Line 321-324: It would look nice to add the harmful effects of mules loading and transportation here.

Lines 337-341: Do they provide them with feed and water during the journey? And whether during the rest break they unload the animals from vehicles?

Line 350: And what does science say about the space requirement per mule/equid? You can add it here for comparison.

Line 358-361: Was your study population in Nepal following these laws? Re: feed, water, and rest? If yes/no, please state it here.

Line 374: What are the possible causes of these respiratory issues? And how will dehydration affect animal health? This is simple but would look great for a non-technical reader/policymaker/farmer who does not know these things.

Line 390-394: Please state the most common injuries equids may have during the presented journey and how they can be minimized.

Line 423-427: How can load carrying at a very young age badly affect animal physiology and working abilities? For authors’ support:

doi:10.3923/ajava.2016.204.209

doi:10.3389/fvets.2022.886020

doi:10.3389/fvets.2020.00214

Line 482-500: Interesting! Could you please add your recommendations on how the working mule community should acclimatize them before asking the mule to start working in such a situation?

Author Response

(The authors gave the same response as above.)
